# Tomato Maturity Detection and Counting Model Based on MHSA-YOLOv8

**DOI:** 10.3390/s23156701

**Published:** 2023-07-26

**Authors:** Ping Li, Jishu Zheng, Peiyuan Li, Hanwei Long, Mai Li, Lihong Gao

**Affiliations:** Chongqing Academy of Agricultural Sciences, Chongqing 401329, China

**Keywords:** counting, complex environment, maturity grading, tomato, target detection

## Abstract

The online automated maturity grading and counting of tomato fruits has a certain promoting effect on digital supervision of fruit growth status and unmanned precision operations during the planting process. The traditional grading and counting of tomato fruit maturity is mostly done manually, which is time-consuming and laborious work, and its precision depends on the accuracy of human eye observation. The combination of artificial intelligence and machine vision has to some extent solved this problem. In this work, firstly, a digital camera is used to obtain tomato fruit image datasets, taking into account factors such as occlusion and external light interference. Secondly, based on the tomato maturity grading task requirements, the MHSA attention mechanism is adopted to improve YOLOv8’s backbone to enhance the network’s ability to extract diverse features. The Precision, Recall, F1-score, and mAP50 of the tomato fruit maturity grading model constructed based on MHSA-YOLOv8 were 0.806, 0.807, 0.806, and 0.864, respectively, which improved the performance of the model with a slight increase in model size. Finally, thanks to the excellent performance of MHSA-YOLOv8, the Precision, Recall, F1-score, and mAP50 of the constructed counting models were 0.990, 0.960, 0.975, and 0.916, respectively. The tomato maturity grading and counting model constructed in this study is not only suitable for online detection but also for offline detection, which greatly helps to improve the harvesting and grading efficiency of tomato growers. The main innovations of this study are summarized as follows: (1) a tomato maturity grading and counting dataset collected from actual production scenarios was constructed; (2) considering the complexity of the environment, this study proposes a new object detection method, MHSA-YOLOv8, and constructs tomato maturity grading models and counting models, respectively; (3) the models constructed in this study are not only suitable for online grading and counting but also for offline grading and counting.

## 1. Introduction

Tomato fruits are rich in nutrients and have a unique flavor. The tomato was originally produced in South America and has been widely cultivated and promoted in the north and south of China [1]. Tomatoes are generally divided into two types of consumption: raw and cooked. Due to their excellent taste, most young people prefer raw food. Mature tomatoes have the best taste when eaten raw and have the highest market demand [2]. However, there are individual differences in the growth status of tomatoes during the planting process. Planters generally grade tomato maturity, pick and estimate yield through manual observation to ensure market supply [3]. This method is labor-intensive and time-consuming and is not suitable for maturity monitoring during large-scale tomato cultivation.

With the development of artificial intelligence and robotics technology, more and more digital technologies are gradually entering the entire agricultural production chain [4]. The overall development trend is to use robots instead of manual labor for agricultural production operations, ultimately achieving unmanned operations [5]. Taking tomato picking as an example, assuming that robots replace humans in picking mature tomatoes, the robots should first have the same function as human eyes, which is to accurately determine whether there are mature tomatoes in the field of view and where they are located [6]. This function determines whether robots can replace manual picking operations.

In addition, in the actual process of tomato picking and sales, tomatoes are generally divided into three categories, namely mature, semi-mature, and immature [7]. During the sales process, mature tomatoes have relatively soft flesh and are not suitable for remote transportation [8]. Generally, mature tomatoes are sold to nearby shopping malls or supermarkets. Semi-mature tomatoes have a harder flesh and can achieve complete ripening through ripening, making them generally suitable for sale in distant cities [9]. The grading of mature and semi-mature tomatoes is completed through the human eye evaluation of the picking personnel, and there are certain human eye errors in this process [10].

The development of artificial intelligence technology has provided certain technical support for solving this problem [11]. At present, the most mainstream object detection methods have been widely applied in various fields, such as population counting, wheat ear counting, fruit counting, etc. The object detection method itself has two functions: detection and classification, which can detect where the target object is and identify which category it belongs to [12]. Common object detection methods are mainly divided into one-stage and two-stage object detection methods. Among them, the one-stage target detection methods include YOLO series and SSD [13], while the two-stage target detection methods include Faster RCNN, etc. [14]. The one-stage object detection method can directly obtain the position of the target box, without the need to generate many candidate boxes like the two-stage object detection method. In practical applications, the one-stage object detection method has higher effectiveness.

In recent years, some researchers have conducted relevant research on fruit maturity detection and recognition, and have achieved various results [15]. In most cases, the detection of fruit maturity is mainly based on the color changes on the surface of the fruit using machine vision methods, with very few using relevant electronic sensors for detection [16,17,18,19,20,21]. For example, [22] used principal component analysis (PCA) and linear discriminant analysis (LDA) to classify the maturity of cherry tomatoes, with an overall accuracy of 94.4%. The authors of [23] compared the performance of hue-mean and red-green mean in tomato maturity recognition and found that the red-green mean has a better effect, with recognition accuracy of over 96%. In another study, ref. [24] used PCA, Support Vector Machine (SVM), and LDA to classify the color features of tomatoes, and the results showed that the maturity classification model based on SVM had an accuracy of 90.8%. The authors of [25] used threshold segmentation methods to observe the color of tomato leaves and fungal infection to distinguish tomato maturity. The authors of [26] estimated the maturity of tomatoes using partial least squares (PLS) by combining visible/near-infrared spectroscopy and machine vision information, with a model recognition accuracy of 90.67%. In terms of fruit maturity detection for other crops, [27] proposed an automatic classification method for persimmons based on fruit appearance color discrimination maturity. By combining LDA and Quadratic Discriminant Analysis (QDA) to classify the color features of fruits, the overall accuracy can reach 90.24%. Another study [28] used MATLAB to extract the shape, size, and color features of mango fruits, and built an automated analysis tool for mango fruit maturity. The authors of [29] also conducted similar work. The authors of [30] collected image data of apricots, preprocessed them with cropping, filtering, and segmentation, and then extracted image features. LDA and QDA classifiers were used to classify the maturity of apricots, with accuracy of 0.904 and 0.923, respectively. The authors of [31] classified the maturity of dragon fruit by converting RGB image to HSV image and combining naïve Bayes method, and the accuracy rate reached 86.6%. Another study [32] collected relevant image data for *Lycium barbarum* L. and conducted preprocessing such as binarization and morphology, then established a maturity classification model using SVM, with an accuracy of 100%. Similar studies using traditional image processing methods for fruit maturity detection include [33].

With the development of deep learning, the extraction of image features has been automated, and the construction of classification models has become increasingly simple [34,35,36]. The authors of [37] constructed a maturity detection model for tomatoes based on an improved DenseNet, with a detection rate of up to 91.26%. The authors of [38] constructed a tomato fruit maturity detection model using YOLOv4, with an accuracy of up to 95%. The authors of [39] used RCNN to identify, locate, and measure the mature fruits of tomatoes, with a precision of over 95%. The authors of [40] constructed a classification model for avocado ripeness based on CNN, with an accuracy of 93%. Another study [41] collected image data of passion fruit based on Kinect depth camera, used Faster RCNN to detect the fruit, and then used Dense Scale Invariant Feature Transform (DSIFT) combined with Position Locality-constrained Linear Coding (LLC) to extract features from each detection box area. The extracted features were then input into SVM for maturity classification model construction, with a classification accuracy of 91.52%. The authors of [42] constructed a classification model for blueberry maturity using YOLOv3-spp, and achieved a Recall of 91%. The authors of [43] demonstrated the best recognition precision of YOLOv4 for blueberry maturity detection, at 88.12%. YOLOv4 tiny has the fastest inference speed, with an inference time of only 7.8 ms. The authors of [44] used Mask-RCNN to identify the maturity of pineapple, with an mAP of 86.7%. Another study [45] first detected citrus fruits using YOLOv5, and then used ResNet34 to classify the maturity of citrus fruits using significance maps, with an accuracy of 95.07%. The authors of [46] proposed an improved version of the YOLOv4 grape maturity detection and localization model, SM-YOLOv4, with an average accuracy of 93.52%. The authors of [47] improved YOLOv5 using GhostNet and CBAM attention mechanisms and proposed a new method, GGSC YOLOv5. Based on this method, the maturity detection of Hemerocallis citrina Baroni was performed with a detection accuracy of 84.9%. In addition, relevant studies have proved that transfer learning can help improve the reliability of tomato maturity detection model [48]. The authors of [49] proposed a cherry maturity detection model based on YOLOX-EIoU-CBAM, with an mAP of 81.1%.

Although previous studies have used traditional image processing and deep learning methods to research fruit maturity and achieved fruitful results, there are still areas for improvement. (1) Although traditional image processing methods require small data volume and fast computation speed, they have certain scene specificity and may not achieve satisfactory recognition results in complex actual production scenarios. (2) Some studies aim to establish an offline tomato maturity grading system, mainly focusing on the construction of maturity grading models for individual tomatoes under a single background, which is not suitable for the online maturity grading requirements of group tomatoes in complex planting environments. (3) Some studies mainly focus on the recognition of mature fruits, and fruit maturity classification cannot meet actual production needs. (4) Some studies mainly focus on model construction from a single task perspective, without considering multi-task model construction.

Therefore, this study focuses on the maturity grading and counting requirements that exist in the actual planting and sales process of tomatoes. Tomato image datasets with factors such as occlusion and light interference are collected in the actual planting environment. Based on the current state-of-the-art one-stage object detection method YOLOv8 [50], a multi-head self-attention mechanism (MHSA) is used to enhance the backbone end features of YOLOv8 [51], ensuring the effectiveness of feature extraction in the model, A new object detection method MHSA-YOLOv8 has been proposed. Then, this method is used to construct tomato maturity grading and counting models from the perspectives of maturity grading and counting, laying a good foundation for unmanned operations during tomato planting. The specific contributions are as follows:
(1)A tomato maturity grading and counting dataset consisting of three categories and one category was constructed, taking into account practical challenges such as external light interference and occlusion.(2)A target detection method based on MHSA-YOLOv8 has been proposed, further improving the model detection performance of YOLOv8.(3)From the perspective of multi-task model construction, tomato maturity grading and counting models were constructed, respectively, providing technical support for unmanned operations during tomato planting.

## 2. Materials and Methods

### 2.1. Data Acquisition

The data needed for modeling in this study were collected from Shouguang Smart Agricultural Science and Technology Park in Shandong Province, China. With Provence tomatoes as the research object and RGB cameras as the main data collection tool, tomato fruit datasets in different growth states were collected in the real production scene. To ensure the universality and robustness of the model, occlusion, shadows, and light interference are fully considered during the data collection process, ensuring the reliability of the model from the source of the data. The specific data collection example is shown in Figure 1.

### 2.2. Data Preprocessing

This study collected a total of 2208 tomato image datasets in an actual production environment. The entire dataset was divided into training, validation, and testing sets in a 7:1:2 ratio. The specific data distribution is shown in Figure 2.

This study used LabelImg software (1.8.3) to annotate the collected image dataset. Among them, due to the fact that this study constructed models from the perspectives of tomato maturity grading and counting, there were slight differences in the data annotation process. For the tomato maturity grading model, three types of labels, immature (IM), semi-mature (SM), and mature (M), are used for dataset labeling. For the tomato counting model, “tomato” is used as the label for the entire dataset. The specific annotation example is shown in Figure 3.

### 2.3. MHSA-YOLOV8

In the actual tomato planting process, the supervision of tomato maturity and yield is one of the main tasks for planting personnel. In large-scale planting scenarios, the workload of this task is relatively large. Therefore, an intelligent means is needed to achieve real-time maturity grading and counting of tomatoes. However, the actual planting environment is complex and the background interference is significant. In order to further improve the precision and reliability of intelligent means, this study proposes an improved YOLOv8 object detection method based on MHSA, which can be used not only to construct tomato maturity grading models, but also to construct tomato counting models. The specific network structure is shown in Figure 4.

The MHSA-YOLOv8 object detection network structure mainly consists of three parts, namely Backbone, Neck, and Head. Among them, basic data preprocessing operations, including data enhancement, are performed before inputting image data into Backbone. The main function of Backbone is to extract feature information of the target area in the input image. When the image data are input into Backbone, the target areal features are extracted through Conv, C2f and SPPF in turn, and then the obtained features are further processed through the MHSA attention mechanism module to increase the feature weight of the target area, so as to extract more effective feature information. The main function of Neck is feature fusion. From Figure 4, it can be observed that there are three different scale network branches in the backbone input to the Neck part, including the backbone end feature branches after feature enhancement using MHSA. The three feature branches after feature fusion of Neck are input into the Head part again to classify and detect the target areal feature. The main output information here includes the location information and category information of tomato fruit. The difference between the tomato maturity grading model and the counting model is that the former outputs multi-category information in the Head part, while the latter only outputs single category information identifying tomato targets in the Head part. The specific introduction of each module is as follows.

#### 2.3.1. Model Input

In the construction process of tomato maturity grading model and tomato counting model in this study, the input image size was 640 × 640. When inputting image data, MHSA-YOLOv8 adopts the same data enhancement strategy as YOLOv5.

#### 2.3.2. Backbone

The backbone of MHSA-YOLOv8 adopts the Darknet-53 network and replaces all C3 modules in the backbone with C2f modules. Compared to the C3 module, the C2f module is designed by referencing the ideas of the C3 module and ELAN, which can extract richer gradient flow information while maintaining lightweight. In addition, C2f has added more jumper connections and split operations. The number of blocks in C2f has been modified from 3-6-9-3 to 3-6-6-3.

In the tomato maturity grading model involved in this study, some categories have certain similarity due to external light environment interference. In order to ensure that the model extracts richer feature information, this study uses the MHSA attention mechanism to enhance the feature branches output by the backbone end. Usually, a typical self-attention mechanism consists of three matrix operations: Q, K, and V, which essentially belong to self-operation. MHSA is an upgraded version of the typical self-attention mechanism, where each attention operation can extract effective feature information from multiple dimensions through grouping, as shown in Figure 5.

#### 2.3.3. Neck

The Neck part still adopts the SPP-PAN structure. Unlike YOLOv5 and YOLOv6, MHSA YOLOv8 replaces the C3 module and RepBlock of the Neck with C2f, removing the 1 × 1 convolution before up-sampling and directly inputting the feature maps output from different stages of the backbone into the up-sampling stage.

#### 2.3.4. Head

The Head part of MHSA-YOLOv8 has two main improvements compared to YOLOv5: (1) adopting the current mainstream decoupling head structure to separate classification and detection; (2) replacing Anchor-based with Anchor-free.

#### 2.3.5. Network Structure Parameters

In order to facilitate readers to have a clear understanding of the network structure and parameter information of MHSA-YOLOv8, we have listed relevant information as shown in Table 1.

### 2.4. Model Train and Evaluation

The training process of the tomato maturity grading model and tomato counting model is based on RTX3090 (24G), and the dataset partitioning ratio used by the two models is consistent. The detailed training process parameter settings are shown in Table 2.

The tomato maturity grading model and tomato counting model constructed in this study belong to the problem of target detection. After the model construction is completed, Precision, Recall, F1-score, and mAP50 are used as the main model validation indicators for model performance evaluation [46,50]. Among them, AP is Average Precision. The specific calculation formula for the validation indicators is as follows:(1)$$\mathrm{Pr}ecision=\frac{TP}{TP+FP}\times 100\%$$
(2)Recall=TPTP+FN
(3)F1−score=2×Recall×PrecisionRecall+Precision
(4)$$mAP=\frac{\sum {}_{i=1}^{C}AP_{i}}{C}$$
where TP, FP, FN, and TN represent true positive, false positive, false negative, and true negative, respectively. C represents the total number of categories, and AP_i_ represents the AP value of the i-th category.

## 3. Experimental Results

### 3.1. Comparison of Modeling Results of Classical Object Detection Methods

In order to better monitor the ripeness and yield of tomato fruits during the cultivation process online, this study adopts target detection methods to construct tomato ripeness grading models and counting models. Maturity grading is a multi-classification detection task, while counting is a single classification detection task. Compared to single classification detection tasks, modeling multiple classification tasks is more challenging. Therefore, this study first compares and analyzes multiple classic methods based on the tomato maturity grading dataset, in order to obtain a relatively optimal modeling method. The specific modeling results are compared as shown in Table 3.

As we can see from Table 3, YOLOv8 has shown significant advantages over other methods in the construction of tomato maturity grading models. On the premise of prioritizing model performance over other methods, the model size is only 21.4 M, second only to the model size of YOLOv5. Therefore, this study will improve the subsequent model based on YOLOv8.

### 3.2. Modeling Results of Tomato Maturity Detection Model Based on MHSA-YOLOV8

In Section 3.1, we conducted a comparative analysis of modeling based on the tomato maturity grading dataset. Although YOLOv8 has better modeling advantages compared to other methods, the performance of the model still needs further improvement. MHSA is a multi-head attention mechanism that to some extent alleviates the problem of poor modeling ability of single head attention mechanisms in limited feature subspaces. This study further enhances the model’s ability to extract feature diversity and enhance the performance of the tomato maturity grading model by adding MHSA to the end of the YOLOv8 backbone feature extraction network. The specific comparison results are shown in Table 4.

As we can see from Table 4, the performance of the YOLOv8 model improved by MHSA has been further improved. Overall, MHSA-YOLOv8 has improved on Recall, F1-score, and mAP50 by 0.044, 0.003, and 0.004 compared to YOLOv8. In terms of model classification performance for each category, there is a certain degree of feature similarity between SM and M due to external light interference. Therefore, the overall recognition performance of the model in IM categories is better than that of SM and M. Through comparison, it was found that MHSA-YOLOv8 has a certain performance improvement in the recognition reliability of each category, which further proves the effectiveness of MHSA in extracting diversity features.

In addition, due to the relatively complex dataset of tomato maturity grading collected in this study, overfitting of the model is the biggest concern during the training process. Therefore, this study plotted the relevant curves during the training and validation process, and the specific results are shown in Figure 6.

As we can see from Figure 6, during the training and validation process, the loss curve showed a trend of first rapid decline and then gradually flattening, while the curves of validation indicators such as Precision, Recall, and mAP showed a trend of first rapid rise and then tending to flatten and stabilize. It has been proven that there is no overfitting problem in the process of constructing tomato maturity grading models based on MHSA-YOLOv8, and the model exhibits good convergence.

In order to more intuitively evaluate the actual effect of MHSA-YOLOv8 tomato fruit maturity grading model in the recognition of various categories, this study has drawn the confusion matrix of MHSA-YOLOv8 model, as shown in Figure 7.

As we can see from Figure 7, the tomato maturity grading model based on MHSA-YOLOv8 has the best recognition performance in IM categories, followed by M. Due to the fact that SM is a growth stage between IM and M, interference from external light environments can lead to similarity in color features between SM and M, resulting in a certain error rate in the model’s judgment of SM and M.

### 3.3. Modeling Results of Tomato Counting Model Based on MHSA-YOLOv8

The tomato counting model can provide some data support for yield estimation. In Section 3.2, the effectiveness of MHSA-YOLOv8 has been demonstrated based on the tomato maturity grading dataset. In this section, a tomato counting model was constructed using MHSA-YOLOv8, and the specific results are shown in Figure 8.

As we can see from Figure 8, compared with the tomato maturity grading model, the validation indicators of the tomato counting model can achieve relatively ideal results. The reason for this is that the modeling difficulty of single category detection models is lower than that of multi-category detection models, and the performance of the models is relatively good. There are three categories of datasets in the tomato maturity grading dataset, among which SM and M have high similarity due to external light interference. In addition, the number of instances for each category in the entire dataset is also uneven, resulting in limited performance during model training. The tomato counting dataset is a dataset that unifies the three categories of the former as “tomato”; the entire dataset only contains one type of instance and has a large number. During the training process of the counting model, all three categories of tomato fruits are considered as one category for feature information extraction, and the MHSA attention mechanism can improve the backbone’s ability to extract effective feature information of tomato targets in the target area, so the effectiveness of the constructed counting model is better than that of the maturity grading model.

Similarly, to demonstrate the reliability of the model training process, this study plotted the relevant curves during the training and validation process of the counting model, as shown in Figure 9.

As we can see from Figure 9, we can observe that the relevant curves exhibit good convergence, which proves the reliability of the tomato counting model constructed based on MHSA-YOLOv8 in this study.

### 3.4. Practical Application Effect of Tomato Maturity Detection and Counting

In Section 3.2 and Section 3.3, this study used MHSA-YOLOv8 to construct tomato maturity grading models and tomato counting models, respectively. In order to verify the effectiveness of the constructed model in practical applications, this study randomly selected eight images from the test dataset for tomato maturity grading and counting, and the specific results are shown in Figure 10 and Figure 11.

Through comparative analysis of Figure 10 and Figure 11, it was found that both the maturity grading model and the counting model can achieve good model recognition performance within an effective field of view. However, there are still certain limitations to the model’s effectiveness in severe occlusion situations, especially when it comes to maturity grading. In terms of the detection effectiveness of tomato objects, the counting model has shown better performance, especially in the face of partial occlusion. The reason for this may be that the network can learn rich and diverse features during the construction process of a single category model, which has certain advantages in dealing with complex environments.

## 4. Discussion

The main objective of this study is to achieve automated maturity grading and counting during tomato planting, in order to assist the operating robot in achieving online maturity grading and counting, greatly reducing the time-consuming and labor-intensive problem of manual grading and counting. Due to the presence of multiple tomato fruits on a single tomato plant and differences in individual fruit growth. Compared to using traditional image processing methods for tomato maturity grading, object detection methods are more suitable for online grading [41]. For example, some studies have conducted IM, SM, and M grading on cherry tomatoes, but they used a combination of PCA and LDA methods [22,24,25,27]. The model construction process is relatively complex and only suitable for offline grading of individual tomatoes. Although some studies have conducted maturity grading for tomatoes, they mainly focus on individual tomatoes in offline state and do not meet the online grading requirements in actual production processes [23]. Although image classification methods based on convolutional networks can achieve fruit maturity grading, they are generally suitable for individual maturity grading and not suitable for group target grading containing fruits of different maturity levels [9]. Previous studies have demonstrated the feasibility of using object detection methods for fruit maturity grading and counting [13,49]. Some studies have also used YOLO series algorithms for fruit maturity grading, but most have used YOLOv3, YOLOv4, or YOLOv5 [52]. YOLOv8 is the latest object detection method in the current YOLO series, which has better performance advantages compared to earlier versions of YOLO algorithms. Therefore, the best modeling method YOLOv8 was preliminarily selected by comparing mainstream object detection methods in Section 3.1, which is also the most popular one-stage object detection method currently.

However, the actual tomato planting scene is relatively complex, and problems such as leaf obstruction, external light environment interference, and obstruction between fruits pose certain challenges to the maturity grading and counting of tomatoes [3]. In addition, the tomato maturity grading model is more difficult to construct than the tomato counting model. Therefore, this study used MHSA to improve YOLOv8, which to some extent improved the effectiveness of the tomato maturity grading model. Considering the performance improvement of MHSA-YOLOv8 in tomato maturity grading tasks, this study further constructed a tomato counting model based on this method and found that the counting model has better performance improvement compared to the maturity grading model. The difference between maturity grading models and counting models is that the former is a three-classification problem, while the latter is a single-classification problem. In the process of model construction, the single classification problem allows the network to learn rich and diverse feature information of tomato fruits under different growth states, which is more conducive to the model detecting target objects in complex environments. Therefore, the counting model performs better than the maturity grading model.

Whether it is a tomato maturity grading model or a counting model, the main reasons limiting the further improvement of model performance are tomato fruit occlusion, background virtualization, and category similarity issues under external light environment interference outside the effective visual field. In the subsequent research process, if the model is deployed in the mobile robot assisted operation process, maturity grading and counting can be considered only for tomato targets within the effective field of view. Targets outside the effective field of view are generally tomato fruits growing on other ridges, which are not within the scope of operation and can be ignored. Secondly, the maturity grading model and counting model constructed in this study are based on complex growth environment conditions, and the model has certain ability to cope with complex environments. Therefore, if applied to offline tomato grading and counting, it also has great potential. Finally, the reason for constructing an online tomato fruit maturity grading and counting model in this study is to achieve graded packaging while machine picking tomato fruits, in order to avoid physical trauma caused by secondary grading. Therefore, the model constructed in this study will have great potential and advantages in assisting unmanned operating robots in completing graded harvesting tasks in the future. Of course, there are still certain shortcomings in the actual implementation process of this study. If the model constructed in this study is mounted on a mobile robot, the limited area and space that the robot can monitor due to the limited space between the planting slots in actual planting modes will limit the robot’s operational efficiency.

## 5. Conclusions

This study focuses on the digital monitoring requirements during tomato cultivation and constructs an artificial intelligence model that can assist intelligent equipment such as unmanned robots in precise operation from the perspectives of tomato maturity grading and counting. First, a tomato maturity grading and counting dataset derived from actual production scenarios was constructed, providing data support for the construction of maturity grading and counting models. Second, to ensure the reliability of the model in practical complex planting scenarios, this study improves YOLOv8 based on the MHSA attention mechanism and proposes a new object detection method, MHSA-YOLOv8. The mAP of the tomato maturity grading model and counting model constructed based on this method are 0.864 and 0.916, respectively, providing good technical support for precise grading picking and counting of tomato fruits. Finally, the models constructed in this study are applicable to actual production scenarios and can achieve online tomato maturity grading and counting. However, occlusion, background virtualization, and light interference are still the main factors that limit the further improvement of the model performance. In the subsequent research process, it is necessary to try to mitigate the impact of the above confounding on the model performance.

## Figures and Tables

**Figure 1 sensors-23-06701-f001:**
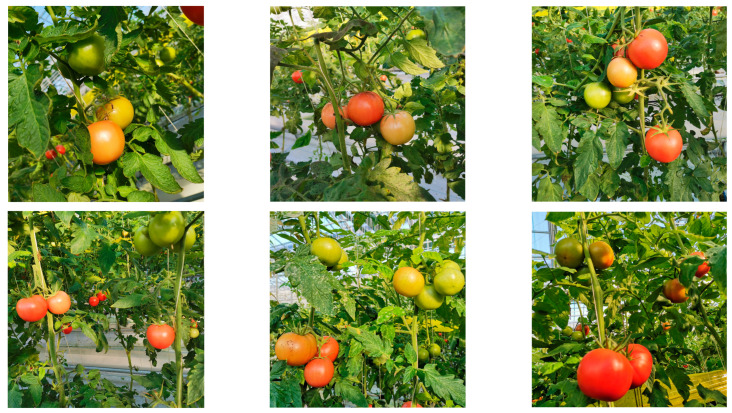
Tomato data collection. This study mainly collected data during the fruit setting period of tomatoes, which is in the third inflorescence growth stage.

**Figure 2 sensors-23-06701-f002:**
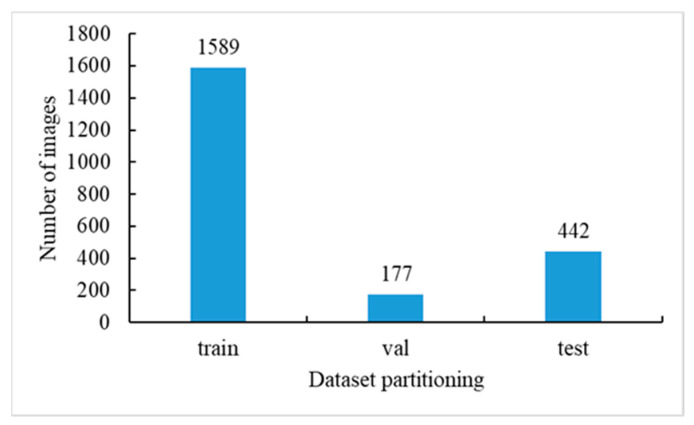
Dataset distribution results. The dataset is mainly divided into train, val, and test, as shown on the horizontal axis. The vertical axis mainly represents the number of images in each dataset.

**Figure 3 sensors-23-06701-f003:**
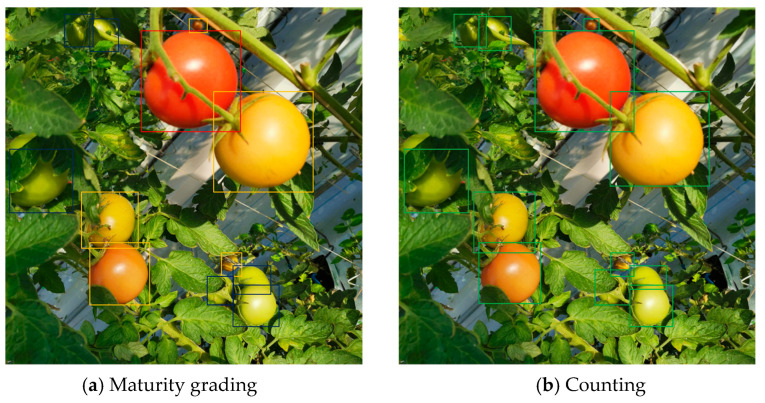
Dataset annotation. Among them, the maturity grading dataset mainly uses three different colored rectangular boxes for annotation during the annotation process, while the counting dataset only needs one colored rectangular box for annotation.

**Figure 4 sensors-23-06701-f004:**
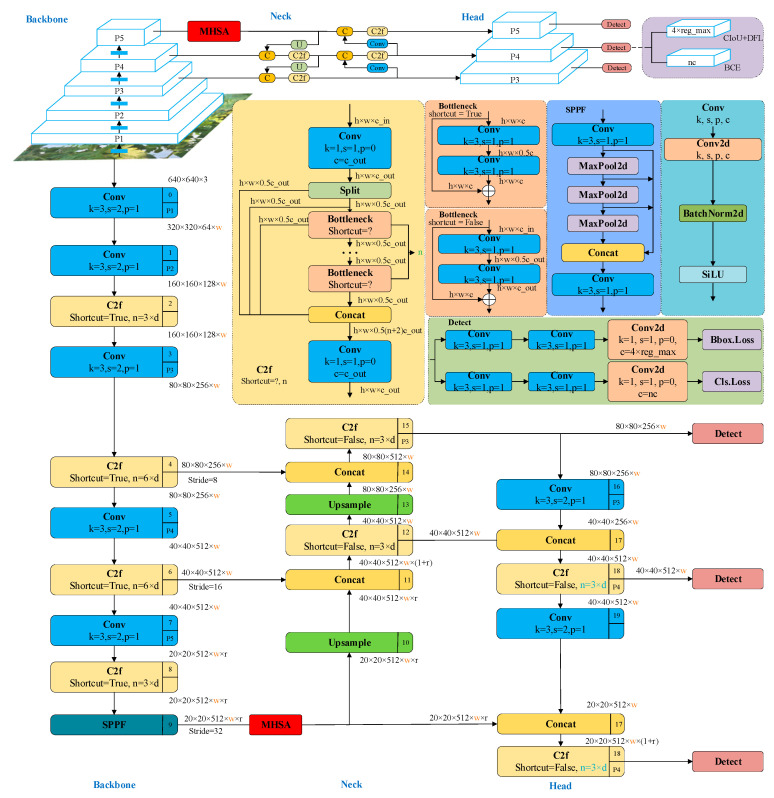
Network structure of MHSA-YOLOV8. Among them, the black arrow represents the direction of data flow during network operations, and different colors represent different network modules. For example, blue represents convolutional modules and yellow represents C2f modules. The name of each module is located at the top left or bottom left of the rectangle where it is located.

**Figure 5 sensors-23-06701-f005:**
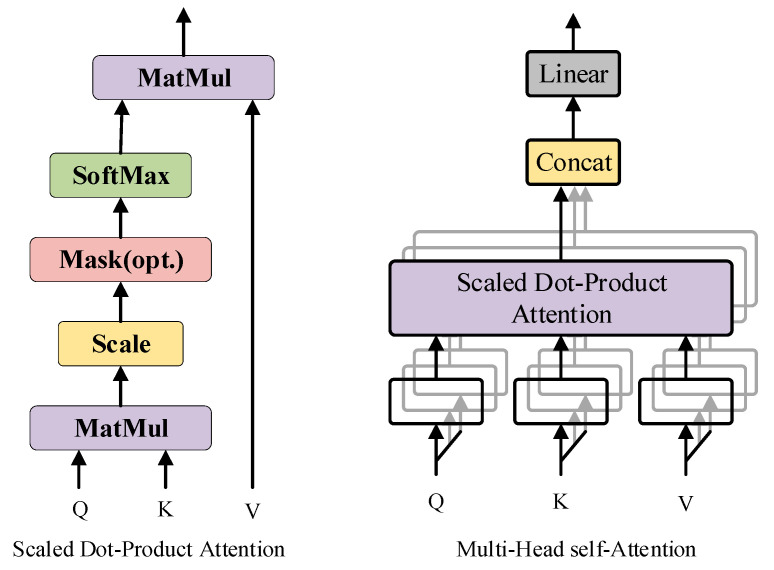
The Network structure of MHSA. Among them, the black arrow represents the direction of the data flow during the calculation process of the attention mechanism, and different colors represent the different modules that make up the attention mechanism.

**Figure 6 sensors-23-06701-f006:**
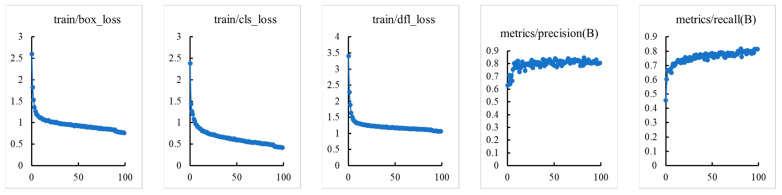
Training and validation process curve of tomato maturity grading model.

**Figure 7 sensors-23-06701-f007:**
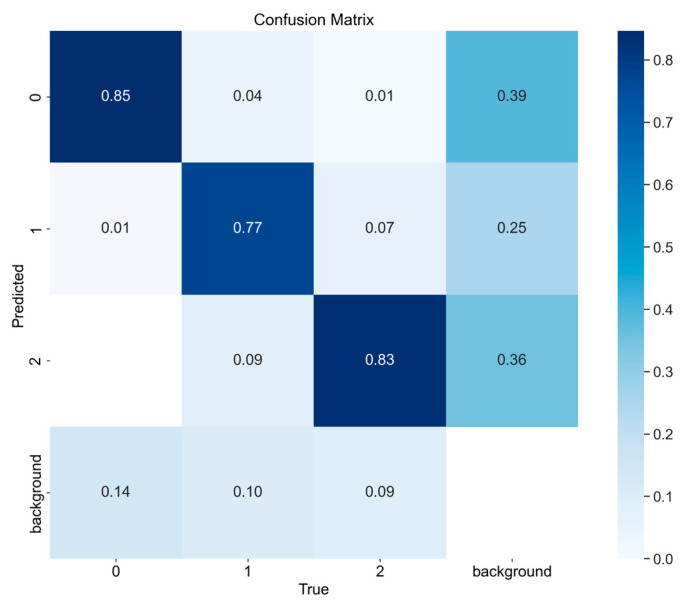
Confusion matrix of tomato maturity grading model.

**Figure 8 sensors-23-06701-f008:**
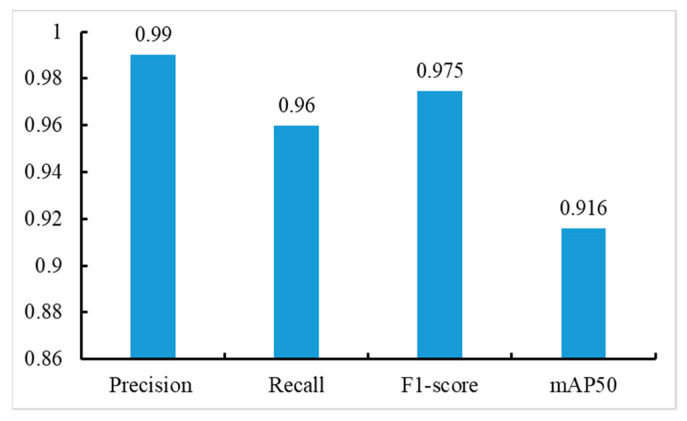
Results of tomato counting model construction.

**Figure 9 sensors-23-06701-f009:**
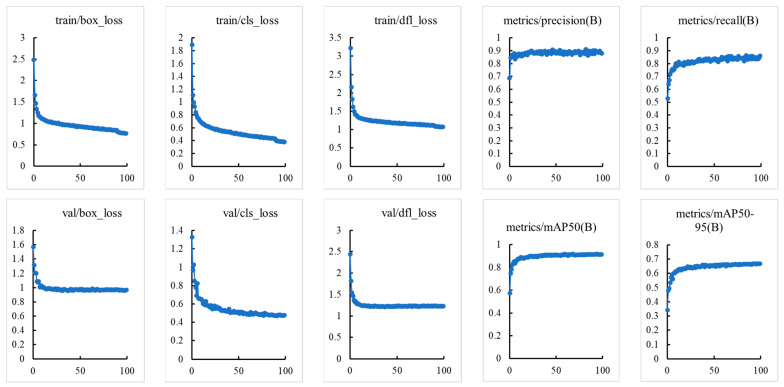
Training and validation process curve of tomato counting model.

**Figure 10 sensors-23-06701-f010:**
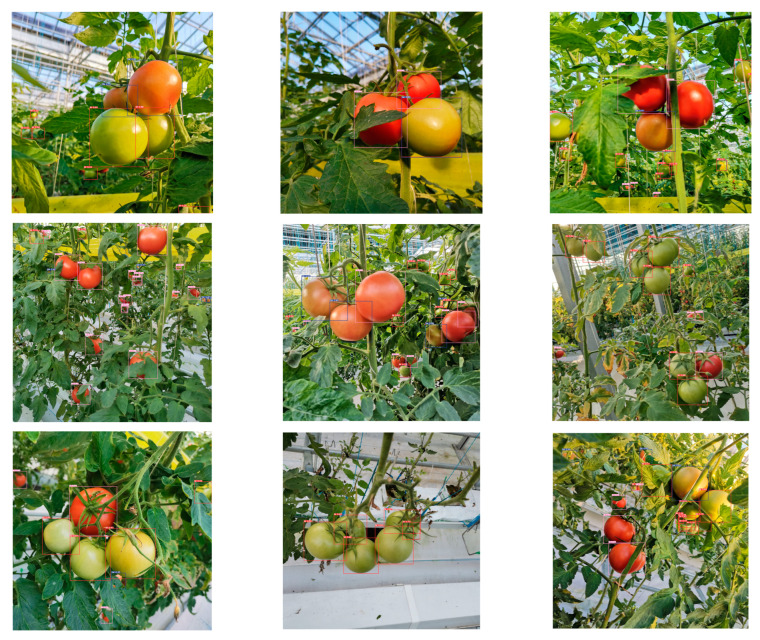
Tomato maturity grading results. The red box represents IM, the blue box represents SM, and the pink box represents M.

**Figure 11 sensors-23-06701-f011:**
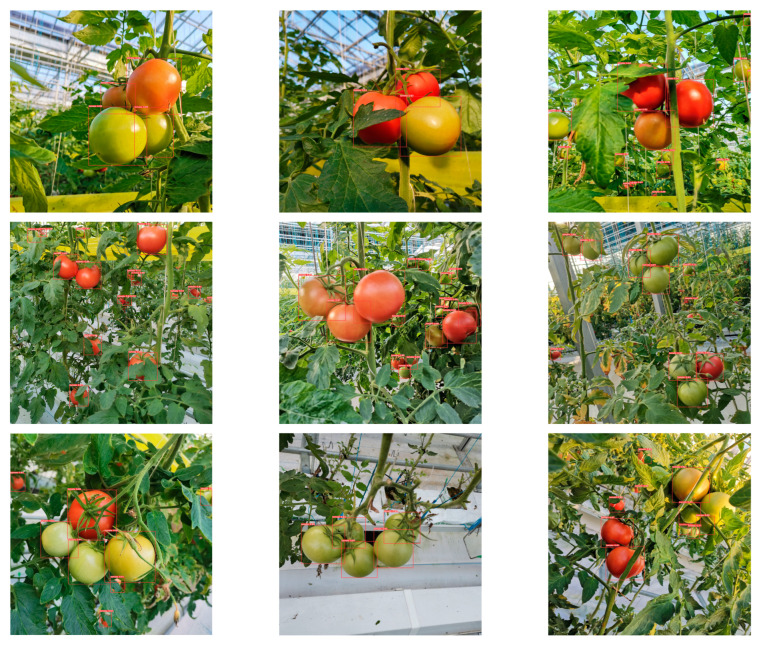
Tomato counting results. The red box represents the location of the tomato target.

**Table 1 sensors-23-06701-t001:** Network Structure and Parameters of MHSA-YOLOv8.

Layers	From	n	Params	Module	Arguments
0	−1	1	928	Conv	[3, 32, 3, 2]
1	−1	1	18,560	Conv	[32, 64, 3, 2]
2	−1	1	29,056	C2f	[64, 64, 1, True]
3	−1	1	79,384	Conv	[64, 128, 3, 2]
4	−1	2	197,632	C2f	[128, 128, 2, True]
5	−1	1	295,424	Conv	[128, 256, 3, 2]
6	−1	2	788,480	C2f	[256, 256, 2, True]
7	−1	1	1,180,672	Conv	[256, 512, 3, 2]
8	−1	1	1,838,080	C2f	[512, 512, 1, True]
9	−1	1	656,896	SPPF	[512, 512, 5]
10	−1	1	787,968	MHSA	[512, 14, 14, 4]
11	−1	1	0	Upsample	[None, 2, ‘nearest’]
12	[−1, 6]	1	0	Concat	[1]
13	−1	1	591,360	C2f	[768, 256, 1]
14	−1	1	0	Upsample	[None, 2, ‘nearest’]
15	[−1, 4]	1	0	Concat	[1]
16	−1	1	148,224	C2f	[384, 128, 1]
17	−1	1	147,712	Conv	[128, 128, 3, 2]
18	[−1, 13]	1	0	Concat	[1]
19	−1	1	493,056	C2f	[384, 256, 1]
20	−1	1	590,336	Conv	[256, 256, 3, 2]
21	[−1, 10]	−1	0	Concat	[1]
22	−1	1	1,969,152	C2f	[768, 512, 1]
23	[16, 19, 22]	1	2,117,209	Detect	[3, [128, 256, 512]]
summary: 230 layers, 11,924,729 parameters, 11,924,713 gradients, 29.3 GFLOPs					

**Table 2 sensors-23-06701-t002:** Parameter settings for model training.

Parameters	Value
Image-size	640 × 640
Epochs	100
Batch-size	8
lr	0.01
Momentum	0.937
Weight_decay	0.0005
warmup_epochs	3
Optimizer	SGD
Loss	VFL_loss, CIOU_loss + DFL

**Table 3 sensors-23-06701-t003:** Comparison of modeling methods.

Methods	Precision	Recall	F1-Score	mAP50	Model-Size
Faster-RCNN	54.0%	0.616	0.576	0.597	108 M
YOLOv3	80.9%	0.761	0.784	0.772	235 M
YOLOv4	81.6%	0.671	0.736	0.768	244 M
YOLOv5	84.0%	0.684	0.754	0.778	14.1 M
YOLOv7	74.7%	0.751	0.749	0.798	72 M
YOLOv8	84.7%	0.763	0.803	0.859	21.4 M

**Table 4 sensors-23-06701-t004:** Comparison of effects before and after model improvement.

Methods	Class	Precision	Recall	F1-Score	mAP50	Model-Size
YOLOv8	all	84.7%	0.763	0.803	0.859	21.4 M
	IM	89.4%	0.779	0.833	0.878	
	SM	80.8%	0.702	0.751	0.815	
	M	83.9%	0.809	0.824	0.883	
MHSA-YOLOv8	all	80.6%	0.807	0.806	0.864	22.9 M
	IM	84.6%	0.829	0.837	0.88	
	SM	76.4%	0.764	0.764	0.824	
	M	80.7%	0.828	0.817	0.888	

## Data Availability

Not applicable.

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
