# Peer review of "Tomato Maturity Detection and Counting Model Based on MHSA-YOLOv8"

_sensors, 2023, doi:10.3390/s23156701_

Round 1

Reviewer 1 Report

This manuscript (sensors-2502458) employs the MHSA-YOLOv8 mechanism, an intersection of artificial intelligence and machine vision, to automate the grading and counting of tomato fruit maturity, overcoming the inaccuracies of traditional manual methods. The technology demonstrated high precision and recall in maturity grading and counting tasks, indicating its potential in supporting both unmanned operations and offline tasks.

The discussion and conclusion sections need to be more detailed. However, it is an interesting study, especially considering the technology and models applied.

Keywords in alphabetic order;

The captions for the figures need better descriptions. For instance, in Figure 1, what is the fruiting time? Which raceme were the images collected from? In Figure 2, describe the x-axis completely or explain the abbreviations in the caption. The same goes for Figure 3, which appears to be a repetition of Figure 1 (as there is no description about what it is). Figures 4 and 5 need to clarify what the colours, legends, and arrows represent. Could you explain this? The text size needs to be increased in Figure 7.

In the discussion, a more thorough comparison with other authors is needed. Add more references for better substantiation. For instance, two references are not sufficient to express the potential of the presented results. How could YOLOv8 improve in relation to other pre-existing models? The conclusion needs review to be more specific.

Extensive, check grammar and spelling in English.

Reviewer 2 Report

In this paper authors consider the development/improvement of a YOLOv8 based on the MHSA attention mechanism for tomato maturity grading and counting model. Though the study is well documented there are some observations that must be further addressed to improve this work.

In the Abstract section:

    The Abstract is clearly presented but it should address the novelty of the proposed system, as well as the main end-users that this system addresses to.

   “The traditional grading and counting of tomato fruit maturity is mostly done manually, which is time-consuming, laborious, and has large human eye observation errors.”…replace with “The traditional grading and counting of tomato fruit maturity are mostly done manually, which is time-consuming and laborious work, and its precision depends on the accuracy of human eye observation.”

In the Introduction section:

·             Authors state that as disadvantage of previous methods “Some studies mainly focus on the recognition of mature fruits, and fruit maturity classification cannot meet actual production needs”. What are those needs they refer to related to their work?

In section “2. Materials and methods

·              Line 167: replace “labelimg software” with “LabelImg software”.

·        Figure 3 should illustrate the “The specific annotation example” but there is no difference between those two images. Any explanation for this?

·          Lines 173-176: I think it is repetitive the idea of “a maturity grading and counting”. It should be rephrase.

·        The Network Structure of MHSA-YOLOV8 is superficially described. Authors should explain better the processing flow in the diagram since it is scattered in some sub-section. They should not assume that everyone reading the article are familiarized with YOLOV8 structure.

·        Formulas for Precision, Recall, F1-score, and mAP50 should be referenced.

In section “3. Experimental Results

·        Line 234: “This study constructs models from two perspectives: maturity grading and counting..” ..repetitive idea.

·        In Table 3 and Table 4 authors should add % for Precision and should specify somewhere in the paper that AP is Average Precision.

·        Line 242: rephrase “As we can see from Table 3, it can be seen that YOLOv8 has shown”.

·        Lines 289-290: what do authors mean by “The tomato maturity grading model can assist breeders in arranging the expected  time and amount of tomato input into the market.”?

·    More explanation should be provided on how the authors achieved 99% Precision in Figure 8.

·        The arrangement of Figure 10 and Figure 11 is not convenient since it does not allow an intuitive comparison (one must scroll to look at an image and then scroll again to find the same one in the next figure).

 My Conclusion

·        This study does not convince me that it can be efficient for larger surfaces cultivated with tomatoes. For point-to-point analysis it proved to be optimum, but authors should address the issue of the size of the geographical area aimed by their studies.

The Quality of English language is good and minor revisions must be performed to improve the paper.

Reviewer 3 Report

The objectives are well defined, fitting in the scopes of the journal

(Tomato maturity detection).The article is written in an appropriate way and theconclusions interesting for the readership.

Round 2

Reviewer 1 Report

I thank the authors for addressing all my comments. I believe the manuscript (sensors-2502458) has significantly improved, as well as the figure captions as I requested. It might be the authors' style, but ideally, the conclusions should be in correct text format rather than bullet points. Please consider rewriting the conclusions. Other than that, I believe it can be accepted for publication after this small adjustment.

Check grammar.
